# Acetaldehyde Enhances Alcohol Sensitivity and Protects against Alcoholism: Evidence from Alcohol Metabolism in Subjects with Variant *ALDH2*2* Gene Allele

**DOI:** 10.3390/biom11081183

**Published:** 2021-08-10

**Authors:** Yi-Chyan Chen, Li-Fang Yang, Ching-Long Lai, Shih-Jiun Yin

**Affiliations:** 1Department of Psychiatry, Taipei Tzu Chi Hospital, Buddhist Tzu Chi Medical Foundation, New Taipei City 231, Taiwan; 0101tiffany@gmail.com; 2Department of Psychiatry, School of Medicine, Tzu Chi University, Hualien 970, Taiwan; 3Department of Nursing and Research Center for Chinese Herbal Medicine, Chang Gung University of Science and Technology, Taoyuan 333, Taiwan; dinolai@mail.cgust.edu.tw; 4Department of Biochemistry, National Defense Medical Center, Taipei 114, Taiwan; yinsj@ndmc.idv.tw

**Keywords:** alcohol metabolism, genetic variation, *ALDH2*2*, ethanol, acetaldehyde, alcohol sensitivity, alcoholism

## Abstract

Alcoholism is a complex behavior trait influenced by multiple genes as well as by sociocultural factors. Alcohol metabolism is one of the biological determinants that can significantly influence drinking behaviors. Alcohol sensitivity is thought to be a behavioral trait marker for susceptibility to develop alcoholism. The subjective perceptions would be an indicator for the alcohol preference. To investigate alcohol sensitivity for the variants *ADH1B*2* and *ALDH2*2*, sixty healthy young males with different combinatory *ADH1B* and *ALDH2* genotypes, *ADH1B*2/*2–ALDH2*1/*1* (*n* = 23), *ADH1B*2/*2–ALDH2*1/*2* (*n* = 27), and *ADH1B*1/*1–ALDH2*1/*1* (*n* = 10), participated in the study. The subjective perceptions were assessed by a structured scale, and blood ethanol and acetaldehyde were determined by GC and HPLC after an alcohol challenge in two dose sessions (0.3 g/kg or 0.5 g/kg ethanol). The principal findings are (1) dose-dependent increase of blood ethanol concentration, unaffected by *ADH1B* or *ALDH2*; (2) significant build-up of blood acetaldehyde, strikingly influenced by the *ALDH2*2* gene allele and correlated with the dose of ingested alcohol; (3) the increased heart rate and subjective sensations caused by acetaldehyde accumulation in the *ALDH2*2* heterozygotes; (4) no significant effect of *ADH1B* polymorphism in alcohol metabolism or producing the psychological responses. The study findings provide the evidence of acetaldehyde potentiating the alcohol sensitivity and feedback to self-control the drinking amount. The results indicate that *ALDH2*2* plays a major role for acetaldehyde-related physiological negative responses and prove the genetic protection against development of alcoholism in East Asians.

## 1. Introduction

Alcoholism is one of most common mental disorders and cause devastating medical complications and social negative impacts. Alcohol-related disease morbidity and mortality markedly increased in the youngest age group and adulthood [1]. Problematic drinking is a major public health issue, directly leading to physical dysfunction and shortening life expectancy [2]. Genetic, animal, and clinical studies demonstrated that alcoholism is believed to be a multifactorial, polygenic disorder involving complex gene-to-gene and gene-to-environment interactions. Alcohol metabolism is one of the biological determinants that can significantly influence the drinking behavior and the development of alcoholism [3,4,5,6].

Alcohol dehydrogenase (ADH) and aldehyde dehydrogenase (ALDH) are the key enzymes responsible for ethanol metabolism through first-pass metabolism in the liver [3,7,8,9,10]. Most ethanol elimination occurs by oxidation to acetaldehyde and acetate, catalyzed principally by alcohol dehydrogenase (ADH) and aldehyde dehydrogenase (ALDH). Both enzymes exhibit genetic polymorphism and ethnic variation [11,12,13]. The functional polymorphisms of *ADH* and *ALDH* lead to genetic variations on the alcohol metabolism (pharmacokinetics) and responses to alcohol (pharmacodynamics). The effect of ethanol and its metabolite acetaldehyde are important determinants that significantly influence drinking behavior and vulnerability to alcohol dependence [5,14]. Functional variant alleles of *ADH1B*2* and *ALDH2*2* have been consistently replicated to show protection against developing alcoholism and confined in the Asian population, including Han Chinese, Japanese, and Koreans [15,16,17,18]. The genomic scans and genome-wide association studies have shown that alcoholism has a strong linkage to the chromosome 4q which is *ADH* gene loci [19]. Our previous multiple logistic regression analyses also suggest that *ADH1B*2* and *ALDH2*2* may independently influence the risk for alcoholism [15,20]. However, it was difficult to show the meaningful differences of alcohol metabolism and pharmacological responses in the subjects with diverse *ADH1B* gene alleles, even well controlling the *ALDH2* genotypes [21].

In the alcohol challenge studies, the subjects with ALDH2 enzyme deficiency manifest elevated levels of acetaldehyde in blood, as well as facial flushing and tachycardia [21,22,23,24]. This alcohol-induced sensitivity reaction is very similar to the aversive reaction caused by alcohol ingestion in patients being treated with the ALDH inhibitor disulfiram [25]. Preclinical research also suggested the extraction of Kudzu herbal roots which showed to suppress the alcohol consumption by selective inhibition of the ALDH2 enzyme and indicated the therapeutic effect on alcoholism [26,27]. *ALDH2*2* polymorphism has also been associated with alcohol-related diseases, including diabetes, as well as oropharyngeal and esophageal cancer [28]. The enzyme deficiency of *ALDH2*2* gene allele has been found to contribute to the variability of activation and efficacy of nitroglycerin in the treatment of angina and heart failure [29,30]. The findings of ischemia-induced cardiac damage were also proved and rescued by ALDH2 activator in an animal study [31].

Alcohol drinking is moderated by many factors encompassed with social, psychological, and biological components which influence the alcohol exposure, positive/negative reinforcement, and responses. Until now, alcohol sensitivity is thought to be a behavioral trait marker for the susceptibility to develop alcoholism [32]. Longitudinal cohort family studies have demonstrated that subjects with lower levels of alcohol responses have an increased risk of developing alcoholism, because the feedback signals to stop drinking are diminishing. Alcohol sensitivities would reflect multiple phenotypes related to how subjects react to the alcohol challenge or retrospective studies report to the drinking experiences. The subjective perceptions to alcohol represent the balance of positive and negative effects which may be acquired or are familial in nature [33,34]. Numerous studies show that some genetic variants, such as genes related to alcohol metabolism, stress hormone, GABA, opioid, glutamate, and dopamine neurotransmissions, contribute to the alcohol intensity and pharmacological variations [10,35,36]. The differentiation of alcohol sensitivity may provide a suitable way to investigate the candidate genetic factors for the development of alcoholism [32,37]. However, until now, it is difficult to find the consistent conclusions between the phenotypic and genotypic research in the Caucasian population.

The subjective perceptions to alcohol represent the balance of positive and negative effects and will be an indicator for alcohol preference. Considering the ethnic variations in East Asians, the variant gene types of *ADH1B* and *ALDH2* would provide the evidence to prove the concept and to discriminate the pharmacokinetic and pharmacodynamic dual effects, including ethanol and acetaldehyde, on alcohol sensitivity [9,38]. Regarding the alcohol sensitivity and preference, there was still lack of systemic research in East Asian populations. Our previous report focused on the responses of cardiac hemodynamic parameters and cranial arterial blood flow, as the setting of the hospital examination room and intermittent measurements of cardiovascular function could influence the emotional state of the participants. The weakness of our previous study was not naturalistic [9,38]. To address these weaknesses and to prove the preliminary investigation, we separately designed a very comprehensive measure of the alcohol perception and created a new Alcohol Response Scale in this study. To explore the dual pharmacodynamic response of alcohol metabolism, we measured blood ethanol and acetaldehyde and assessed the subjective psychological reactions following the low and high alcohol doses in nonalcoholic subjects with different combinatory *ADH1B* and *ALDH2* genotypes.

## 2. Materials and Methods

### 2.1. Subjects

Considering the gender differences in alcohol metabolism, only male subjects were recruited in this study. The participants were recruited for the study from a total of 545 medical college male students who had been genotyped at the *ADH1B*, *ADH1C*, and *ALDH2* gene loci [15]. There were relatively low frequencies of some genotypes in Han Chinese, ex. 4% of *ADH1B*1/*1* and 1% of *ADH1C*2/*2*. Of these subjects, only individuals with *ADH1B* and *ALDH2* combinatory genotypes were selected for this study. The experimental procedures were approved by the Institutional Review Board for Human Studies, and informed content was obtained from each subject after the nature and the possible consequences of participation in the study had been explained. All participants received the structured interview to screen for alcohol drinking, substance use, as well as psychiatric and medical disorder. Participants who had a history of mental disorder, problematic drinking, alcohol use disorder, hepatitis, and medical disorders were excluded from the study. Sixty male students participated in the study with the following combinatory genotypes: *ADH1B*2/*2–ALDH2*1/*1* (*n* = 23), *ADH1B*2/*2–ALDH2*1/*2* (*n* = 27), and *ADH1B*1/*1–ALDH2*1/*1* (*n* = 10). The participants lived on-campus at the National Defense Medical Center which provided room and board. They were off-campus during weekends and summer/winter breaks, so they had a relatively uniform lifestyle and nutritional status. None of the subjects had a family history of alcohol use disorder. None of the subjects used drugs.

### 2.2. General Procedure

All participants were requested not to consume alcohol for at least 1 week. A partly corroborated blank level for both blood ethanol and acetaldehyde was examined before the alcohol challenge test. Before the study day, the participants received the Alcohol Response Scale (described as below) to familiarize themselves with the structures of the questionnaire and to indicate their past drinking experience and their expected reactions. On the day of testing, the subjects had a light lunch before 11:30 and arrived at the testing site at 14:00. After measuring body weight and height, the subjects relaxed in a relatively quiet and comfortable place for 30 min. Baseline subjective sensation assessment, heart rate, and blood pressure were carried out and blood was drawn for baseline ethanol and acetaldehyde measurements. Then, ethanol (0.3 g/kg) was orally administered within 10 min, i.e., with 2-min interval for a total of 5 aliquots. Taiwan rose red wine (TTL Corp, Taipei, Taiwan, 10.5%, *v*/*v*) was adapted through determination of ethanol concentration (described as below). Heart rate, blood pressure, and self-report of alcohol responses were taken at 10, 25, 40, 55, 70, 85, 100, and 130 min after the start of ethanol ingestion. Blood was collected from the antecubital vein via an indwelling catheter into syringe tubes at 25, 75, and 130 min. To avoid memory learning and alcohol tolerance effect, the test was repeated with a higher dose of ethanol (0.5 g/kg) with a three-week interval. After completion of the alcohol challenge, the participant rested for 1 to 2 h. They showed no apparent discomforting symptoms.

### 2.3. Genotyping

Extraction of DNA from leukocytes and the determination of the *ALDH2* and *ADH1B*, allelotypes, using polymerase chain reaction (PCR)–restriction fragment length polymorphism (RFLP), were carried out as described previously [15].

### 2.4. Determination of Blood Ethanol and Acetaldehyde

The ethanol concentration of Taiwan rose red wine (10.5% *v*/*v*) was determined by headspace gas chromatography through a series of dilution. Blood ethanol concentration was assayed on the study day by gas chromatography (GC, Varian Associates, Sugar Land, TX, USA) as described previously [39]. Blood acetaldehyde was determined by high-performance liquid chromatography (HPLC, Millipore Corp, Burlington, MA, USA), equipped with fluorescence detection, and corrected for blank value as described previously [39].

### 2.5. Assessment of Subjective Response

In this study, a structured self-rating scale, the so-called Alcohol Response Scale (ARS), was adapted to assess alcohol sensitivity and psychological response following alcohol consumption. The rating scale was modified from the Subjective High Assessment Scale [40], Biphasic Alcohol Effect Scale [41], and Body Sensation Scale [42,43]. The scale contained the physiological and psychological components and was divided into two domains of positive and negative sensation to alcohol. The subjective sensations were rated using a 10-division scale, with grades of alcohol intensity ranging from 0, indicating no reaction, to 10, indicating an intolerable range. The rating scale had exceptional construct validity by principal axis analysis, the value of Cronbach’s α was up to 0.9, good reliability, and the correlation value by Spearman test was between 0.4 and 0.8.

During the explanatory session of the experimental protocols before the alcohol testing date, the participants were asked to indicate their past drinking experience and their expected reactions after rapidly consuming two and half bottles of Taiwan beer, which is roughly equivalent to the test dose of 0.5 g/kg for an average man weighing 65 kg. During the test session before consuming alcohol, the participants completed the self-ratings as the baseline data. Subjective responses are expressed as relative rating units, which is rating units obtained at various times after alcohol ingestion minus that of the baseline units.

### 2.6. Statistical Analysis

The difference between the genotypes was evaluated by Student’s *t* test. Paired *t* test was performed to compare the differences of body weight and body mass index between two different dose sessions. Data for each of the genotypic groups at each time point for blood ethanol, acetaldehyde, and percentage change of heart rate (excluding zero value points) appeared to be normally distributed when judged by the Q–Q plot analysis (SPSS, Chicago, IL, USA). The difference between the genotypes was evaluated by analysis of variance (ANOVA) with post hoc Scheffe’s test for data with equal variance or using the Tamhane’s T2 test for those with unequal variance. As the data for subjective responses did not seem to be normally distributed, the nonparametric Mann–Whitney *U*-test was performed for comparison.

## 3. Results

The demographic characteristics of participants, carrying *ADH1B* and *ALDH2* genotypes, recruited in this study are shown in Table 1. The mean age, body weight, and body mass index were similar among three study groups. No significant differences with regard to age, body weight, and body mass index were found between the two session studies by 0.3 and 0.5 g/kg ethanol. There seemed relative homogenous in the study groups for testing the alcohol metabolism and subjective perception.

### 3.1. Blood Ethanol

Figure 1 shows the time course of blood ethanol concentrations after receiving two separate doses of ethanol, 0.3 and 0.5 g/kg. No significant difference in blood ethanol levels was found between the *ALDH2*1/*1* and *ALDH2*1/*2* alcoholic groups at the same ethanol dose ingestion. Regardless to the *ALDH2* genotype, a dose-dependent increase of ethanol concentration was shown following 0.3 and 0.5 g/kg of ethanol. The peak ethanol concentrations at 25 min were 6.3 and 7.4 mM at a dose of 0.3 g/kg as well as 11.8 and 12.7 mM at a dose of 0.5 g/kg, for *ALDH2*1/*1* and *ALDH2*1/*2*, respectively. The 1.7-fold increase of blood ethanol was highly correlated the alcohol challenge dose. There was no significant difference between *ADH1B*1*1* and *ADH1B*2/*2* with the homozygous *ALDH2*1* genotypic group (Appendix A).

### 3.2. Blood Acetaldehyde

Figure 2 shows the time course of blood acetaldehyde following the 0.3 and 0.5 g/kg ethanol ingestion. The *ALDH2* heterozygotic group exhibited strikingly higher blood acetaldehyde levels than those of the homozygotic group during the entire test period. The peak blood acetaldehyde appeared to have a 1.8-fold increase, 46.3 µM vs. 86.0 µM, following the 0.3 and 0.5 g/kg ethanol ingestion, respectively. Regardless of the ethanol ingestion dose, there was no significant increase, 1~2 µM, in the *ALDH2*1/*1* genotypic group. In the same *ALDH2*1/*1* homozygotes, the participants with *ADH1B*2/*2* genotype following the 0.5 g/kg ethanol dose ingestion slightly increased the blood acetaldehyde, around 2.5 µM; however, there was no significant increase for the *ADH1B*1/*1* group regardless of the challenging ethanol dose (Appendix A).

### 3.3. Alcohol Sensitivity and Subjective Response

The basal cardiac hemodynamic parameters, such as heart rate, systolic blood pressure, and diastolic blood pressure before the alcohol challenge were not statistically different between the two combinatory genotypic groups. Following the alcohol ingestion, the heart rates in the heterozygotic *ALDH2*1/*2* group were persistently faster during the entire test period, compared with the homozygotic group (Figure 3). The percent changes of heart rate at 25 min were significant higher in the participants received the 0.5 g/kg ethanol than in those with 0.3 g/kg dose, 43.6 ± 4.1% and 24.8 ± 3.8% (mean ± SE), respectively. There were no heart rate changes in the *ALDH2*1/*1* subjects with a different ethanol dose ingestion.

The subjective perception of palpitation to the alcohol drinking is shown in Figure 4A. The result demonstrated the same trend of the heart rate change and was similar in its rating unit. The higher correlation of up to 0.6−0.8, depending on the time point, was shown between actual heart rate measurement and subjective feelings. It implied good reliability and validity of the Alcohol Response Scale to assess the physiological and psychological feelings after intaking different dose of ethanol. Otherwise, the sensation of facial warming demonstrated a stronger reaction for the subjects with *ALDH2*1/*2* genotypes, but nearly no change to those with homozygous *ALDH2*1/*1* (Figure 4B).

The alterations of dizziness and sedation sensation following ethanol intake are shown in Figure 4C,D. The subjects with *ALDH2* heterozygotes perceived significantly stronger dizziness and sedation effect up to 100 min than the *ALDH2*1* homozygotes. The alcohol sensitivity was escalated following the higher dose of 0.5 g/kg ethanol compared to the low dose, especially dramatic increase in the *ALDH2*1/*2* genotypic group. Consistently, the sensation of drunkenness revealed a same trend and patterns compared with the genotypes and alcohol drinking dose (Figure 5).

The terrible sensation overall suggesting the general preference to alcohol is shown in Figure 6. The participants with *ALDH2*2* heterozygote revealed more negative psychological responses than those with *ALDH2*1/*1* homozygote. There were stronger reactions during the entire test period in the 0.5 g/kg testing dose group. Apart from the physiological negative reactions, a modest increase of positive sensation in the *ALDH2*1/*2* group was noticed; however, they did not show the statistical significance among groups with different alcohol dose ingestion (Figure 6). The *ADH1B*2/*2*–*ALDH2*1/*1* subjects, following the higher ethanol dose, showed slightly elevated physiological negative sensation which may attribute to the modest increase of the blood acetaldehyde level in this study (Appendix A). The two groups of *ADH1B*1/*1* and *ADH1B*2/*2* with *ALDH2*1/*1* homozygote showed no significant changes for positive sensations to alcohol (Appendix A).

## 4. Discussion

In this study, the participants were healthy young male students, matched for age, body weight, BMI, having a psychologically sound mind to respond to the subjective sensations. The homogeneity of our subjects and the structured study procedure provided a good opportunity to reliably report the subjective perceptions following the alcohol challenges. This integrative study clarifies the correlations of pharmacokinetic and pharmacodynamic effects of ethanol and acetaldehyde in subjects with combinatory *ADH1B* and *ALDH2* genotypes by ingesting low and high ethanol doses. The principal findings have shown the following: (1) dose-dependent increase of blood ethanol concentration, unaffected by *ADH1B* or *ALDH2*; (2) significant build-up of blood acetaldehyde, strikingly influenced by the *ALDH2*2* gene allele and correlated with the dose of ingested alcohol; (3) the increased heart rate and subjective sensations caused by acetaldehyde accumulation in the *ALDH2*1/*2* heterozygotes; (4) no significant effect of *ADH1B* polymorphism in alcohol metabolism or producing the psychological responses. To the best of my knowledge, this is the first report to systemically analyze the correlation of pharmacokinetic and dose effects in subjects carrying the different combinatory *ADH1B* and *ALDH2* genotypes.

### 4.1. Pharmacokinetics of Ethanol and Acetaldehyde

The study was created to clarify the dual effect of ethanol and acetaldehyde for the individuals with variant types of *ADH1B* and *ALDH2* and to explore the ethanol dose-related biphasic alcohol responses. The blood ethanol displays a 1.7-fold increase depending on alcohol drinking dose, 0.3 vs. 0.5 g/kg ethanol, regardless of the *ADH1B* and *ALDH2* genetic polymorphisms. The elevation of the blood ethanol level showed highly consistency with our previous studies which were performed in alcohol challenge tests for alcoholic and nonalcoholic subjects with different metabolic genes [24]. Dose-dependent increase of blood ethanol level is demonstrated for the healthy subjects, no matter what the *ADH1B* or *ALDH2*, in this study (Figure 1). The blood acetaldehyde levels increased during the entire testing period in the *ALDH2*1/*2* heterozygotic subjects, but not in the homozygous *ALDH2*1/*1* group. The acetaldehyde accumulation shows a highly positive correlation with testing ethanol dose, ex. peak acetaldehyde increased 1.8-fold, 86.0 µM in 0.5 g/kg testing dose vs. 46.3 µM in 0.3 g/kg ethanol dose, respectively (Figure 2). No significant differences in the blood levels of ethanol and acetaldehyde are found among subjects carrying the homozygous *ADH1B*1/*1* and *ADH1B*2/*2* genotypes regardless the ethanol challenge dose. The study observations strongly indicated the *ALDH2*1/*2* heterozygotes have the inborn enzyme deficiency to remove the blood acetaldehyde during process of alcohol oxidation. The findings are compatible with the protein contents and enzyme kinetic studies of mitochondrial ALDH2 in the human liver. The residual ALDH2 activities in homozygous *ALDH2*1/*1* and heterozygous *ALDH2*1/*2* were markedly reduced down to ~0% and ~16%, respectively, compared to that of the *ALDH2*1/*1* [44]. The residual activity of ALDH2 may contribute to the oxidation of acetaldehyde in individuals with *ALDH2*2* gene allele. Some antialcohol drugs, such as disulfiram and calcium carbamide, were used to treat alcoholism through ALDH enzyme inhibition caused high level of blood acetaldehyde accumulation. In the past 70 years, disulfiram has been widely used to treat alcoholism for the Caucasian population, almost all having the homozygous *ALDH2*1/*1*, and the treatment produced strong aversive alcohol reactions and severe cardiovascular side-effects following alcohol consumption. The pharmacokinetic and pharmacodynamic effects of disulfiram-ethanol reactions appear to have similar mechanisms to alcohol reactions in East Asians with the *ALDH2*2* gene allele.

### 4.2. Alcohol Sensitivity and Subjective Perception

Alcohol sensitivity is a summation pharmacological effect of positive reinforcement and negative aversion. It could imply the alcohol preference and indicate the drinking amount and alcohol tolerability and predict the risk of alcoholism development. So many confounding factors were influenced by alcohol responses, such as sex, age, culture, drinking experiences, psychosocial, and environmental cues. The study recruited relative homogenous subjects with a similar lifestyle and tried to control the bias influenced by the reports to alcohol response. All the experimental procedures were performed under controlled conditions and all participants were blind to the alcohol change dose and their own genotypes. Therefore, the study would be powerful to predict the dose-related alcohol response in subjects with different *ADH1B* and *ALDH2* genetic polymorphism.

Based on our previous studies, heart rate change was the most sensitive indicator of physiological responses during the period of 130 min periods after different alcohol ingestion in the *ALDH2*1/*2* subjects. To test the validity and reliability of the Alcohol Response Scale, there are strong relationships, a correlation coefficient of ~0.63, between the measuring heart rate and subjectively reporting the palpitation sensation during the entire testing session, especially in the *ALDH2*1/*2* group (shown in Figure 3 and Figure 4A). Further regression analysis showed the significant correlation coefficient, up to 0.7, between heart rate change and acetaldehyde level in those with *ALDH2*1/*2* heterozygotes; however, it did not show a difference in the *ALDH2*1/*1* homozygotes. The acetaldehyde demonstrated a dominant effect on the increase of the heart rate, regardless of the blood ethanol concentration. The correlation analyses provide good evidence to prove the reliable questionnaire to assess the psychological responses to the effects of alcohol and acetaldehyde.

The subjective perceptions of facial warming, dizziness, sedation, and drunkenness show good agreement with the measured heart rate changes (shown in Figure 4 and Figure 5). There is a trend of having more heightened physiological perceptions following the higher ethanol dose challenge and showing the more synergistic intense reactions in the *ALDH2*1/*2* subjects. The alterations of heart rate and psychological measurements of dizziness, sedation, and drunkenness appear to be correlated with the blood level of acetaldehyde, rather than that of ethanol. These results indicated that acetaldehyde has the synergistic effect to enhance the alcohol sensitivity and evoke the physiological feedback signals to self-control the drinking amount. The study findings proved the concept of a higher level of alcohol sensitivity to reduce the alcohol consumption and lower the risk for developing alcoholism [32]. The epidemiological research has shown that the individuals with *ALDH2*1/*2* genotype self-reported the lower frequency of alcohol use and heavy drinking behaviors [45,46]. Compatible with genetic association studies, there are genetic variations of alcohol metabolic genes in East Asian populations such as China, Japan, Korea, and Taiwan. The homozygosity of variant *ALDH2*2* gene allele nearly fully protected against alcoholism; however, *ALDH2*1/*2* heterozygosity only had partial protection for developing alcoholism.

### 4.3. Acetaldehyde and Alcoholism

Regarding the results of overall negative and positive sensations to alcohol, the study shows that more terrible feelings were reported in the *ALDH2*1/*2* subjects and intensified by the dose during the 130 min testing period (Figure 6). However, there are no significant differences of great sensations among the groups of *ADH1B*2/*2*–*ALDH2*1/*1* and *ADH1B*2/*2*–*ALDH2*1/*2* following different ethanol doses. Compared to the Wall et al. research [47], the stronger positive and negative alcohol reactions were simultaneously reported for the alcohol flushing subjects of Asian Americans. The inconsistent results may attribute to the different culture and social-familial backgrounds or the sampling variations that some subjects self-reported the alcohol flushing, not precisely representing the *ALDH2*2* allelotype. The discordant finding was difficult to explain the lower frequency of *ALDH2*2* gene allele for alcoholics in the genetic association research. However, it may suggest another viewpoint to search the partial protection against alcoholism on the issue of 15% alcohol dependents with heterozygotic *ALDH2*1/*2* genotype [15].

The current results show that the aversive alcohol sensitivity reactions appear to be correlated with blood acetaldehyde accumulation. Until now, it is an intriguing issue that the subjects with *ALDH2*2* gene allele could override the physiological barrier to develop alcohol addiction. Our previous study has demonstrated that physiological tolerance or innate low sensitivity may play a crucial role in overcoming the aversive reactions for the *ALDH2*1/*2* alcoholics [24]. Comparing the pharmacokinetic profiles between *ALDH2*1/*2* alcoholic and nonalcoholic subjects following the same 0.5 g/kg ethanol dose challenge, the results show similar blood concentration of ethanol and acetaldehyde as well as a significantly higher level of blood acetaldehyde. These results document that the *ALDH2*1/*2* alcoholics could gradually adapt the acetaldehyde aversive effect through repeating drinking caused physiological tolerance, but no apparent metabolic adaptation for the ALDH2 enzyme after chronic alcohol consumption. Regarding the pharmacological basis of acetaldehyde, it is still unclear and an interesting issue. Some animal studies support that in acetaldehyde itself exists a reinforcing effect or to form new products in the brain, such as tetrahydroisoquinolines (THIQ; so called salsolinol), to activate the brain rewarding sensation and to drive the drinking behaviors [48,49]. Thus, acetaldehyde might have dual roles, depending on the location and concentration, to modulate the drinking behaviors in rodents and humans [50,51]. The potential reinforcing effect of acetaldehyde may induce the positive sensation to low amount of alcohol in the *ALDH2*1/*2* subjects that allow other biological and sociocultural factors to facilitate drinking and to overcome the aversive reactions during repetitive alcohol use. It would be susceptible to the development of alcoholism.

## 5. Conclusions

The study explores the correlations of pharmacokinetic and pharmacodynamic effects of ethanol and acetaldehyde in subjects with combinatory *ADH1B* and *ALDH2* genotypes by different ethanol doses. The alterations of subjective perceptions in *ALDH2*1/*2* healthy participants after challenge with different ethanol doses have shown to be primarily caused by the accumulation of acetaldehyde, rather than ethanol. The study findings provide the evidence of acetaldehyde potentiating the alcohol sensitive and feedback to self-control the drinking amount. Regardless of *ADH1B* genetic polymorphisms, the results also indicate that *ALDH2*2* plays a major role for acetaldehyde-related physiological negative responses and prove the genetic protection against the development of alcoholism in East Asians. High alcohol sensitivities and aversive reactions strongly correlated with the acetaldehyde accumulation in the *ALDH2* heterozygotes, the so-called Alcohol Intolerant Syndrome. The genetic mutation of *ALDH2* may imply the evolving protective effects to deter excessive alcohol drinking and prevent the physiological toxic effects. Apart from protection from alcoholism, it is noteworthy that persistent elevated blood acetaldehyde synergistically affects the ethanol-related toxicity and potentially facilitates the progression of organ damage.

## Figures and Tables

**Figure 1 biomolecules-11-01183-f001:**
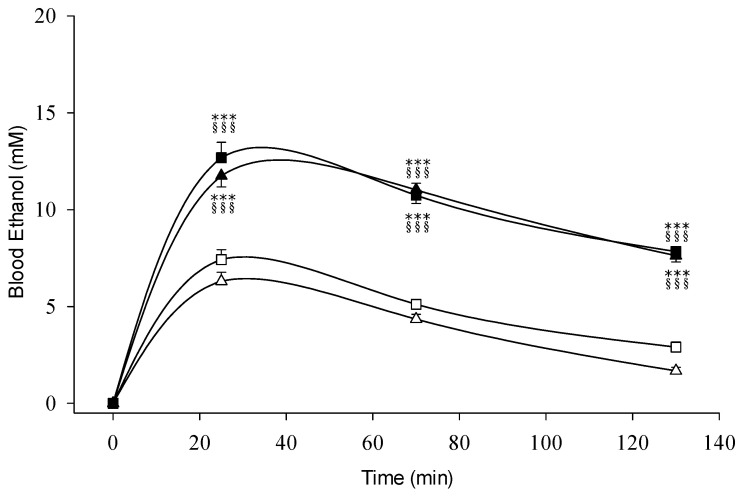
Blood ethanol concentrations of nonalcoholic subjects with different *ALDH2* allelotypes following different doses of ethanol (0.3 g/kg body weight or 0.5 g/kg). All subjects are *ADH1B*2/*2* homozygote. The study groups are denoted as △, *ALDH2*1/*1* with 0.3 g/kg ethanol (*n* = 23); ▲, *ALDH2*1/*1* with 0.5 g/kg ethanol (*n* = 21); □, *ALDH2*1/*2* with 0.3 g/kg ethanol (*n* = 27); ■, *ALDH2*1/*2* with 0.5 g/kg ethanol (*n* = 22). Vertical bars (for clarity only the upper or lower portion shown) represent standard errors of the mean. Statistical differences between groups at the corresponding time points was evaluated by ANOVA. *** *p* < 0.001 vs. *ALDH2*1/*1* with 0.3 g/kg ethanol; ^§§§^
*p* < 0.001 vs. *ALDH2*1/*2* with 0.3 g/kg ethanol.

**Figure 2 biomolecules-11-01183-f002:**
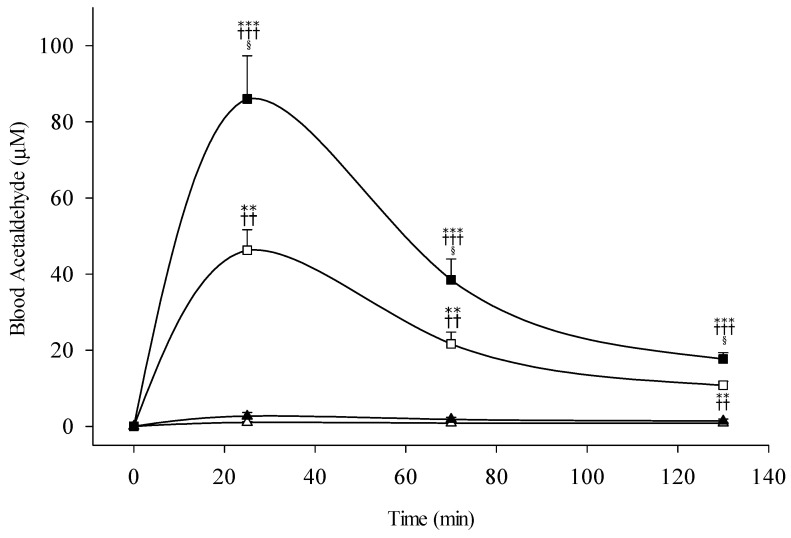
Blood acetaldehyde concentrations of nonalcoholic subjects with different *ALDH2* allelotypes following different doses of ethanol (0.3 g/kg body weight or 0.5 g/kg). All subjects are *ADH1B*2* homozygote. The study groups are denoted as △, *ALDH2*1/*1* with 0.3 g/kg ethanol (*n* = 23); ▲, *ALDH2*1/*1* with 0.5 g/kg ethanol (*n* = 21); □, *ALDH2*1/*2* with 0.3 g/kg ethanol (*n* = 27); ■, *ALDH2*1/*2* with 0.5 g/kg ethanol (*n* = 22). Vertical bars (for clarity only the upper or lower portion shown) represent standard errors of the mean. Statistically significant differences between groups at the corresponding time points was evaluated by ANOVA. *** *p* < 0.001, ** *p* < 0.01, vs. *ALDH2*1/*1* with 0.3 g/kg ethanol; ^+++^
*p* < 0.01, ^++^
*p* < 0.01 vs. *ALDH2*1/*1* with 0.5 g/kg ethanol; ^§^
*p* < 0.05 vs. *ALDH2*1/*2* with 0.3 g/kg ethanol.

**Figure 3 biomolecules-11-01183-f003:**
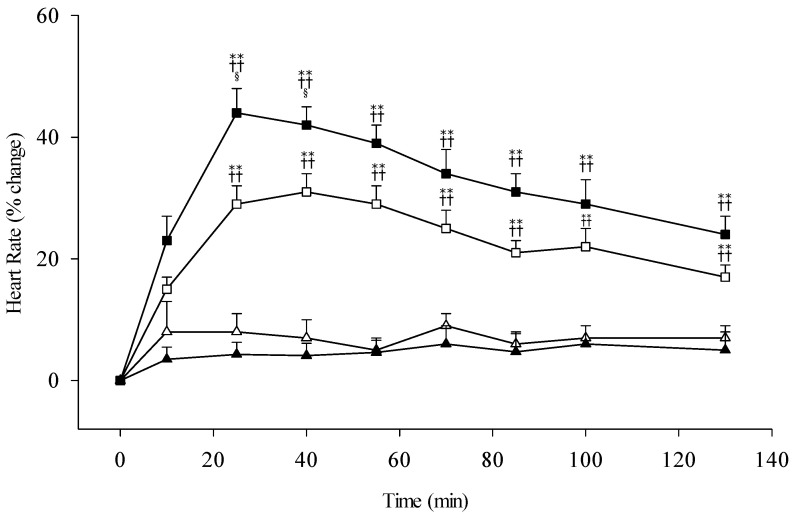
Percentage change of heart rate in subjects with different *ALDH2* allelotypes following different doses of ethanol (0.3 g/kg body weight or 0.5 g/kg). All subjects are *ADH1B*2/*2* homozygote. The study groups are denoted as △, *ALDH2*1/*1* with 0.3 g/kg ethanol (*n* = 23); ▲, *ALDH2*1/*1* with 0.5 g/kg ethanol (*n* = 21); □, *ALDH2*1/*2* with 0.3 g/kg ethanol (*n* = 27); ■, *ALDH2*1/*2* with 0.5 g/kg ethanol (*n* = 22). Vertical bars (for clarity only the upper or lower portion shown) represent standard errors of the mean. Statistically significant differences between groups at the corresponding time points was evaluated by ANOVA. ** *p* < 0.01 vs. *ALDH2*1/*1* with 0.3 g/kg ethanol; ^++^
*p* < 0.01 vs. *ALDH2*1/*1* with 0.5 g/kg ethanol; ^§^
*p* < 0.05 vs. *ALDH2*1/*2* with 0.3 g/kg ethanol.

**Figure 4 biomolecules-11-01183-f004:**
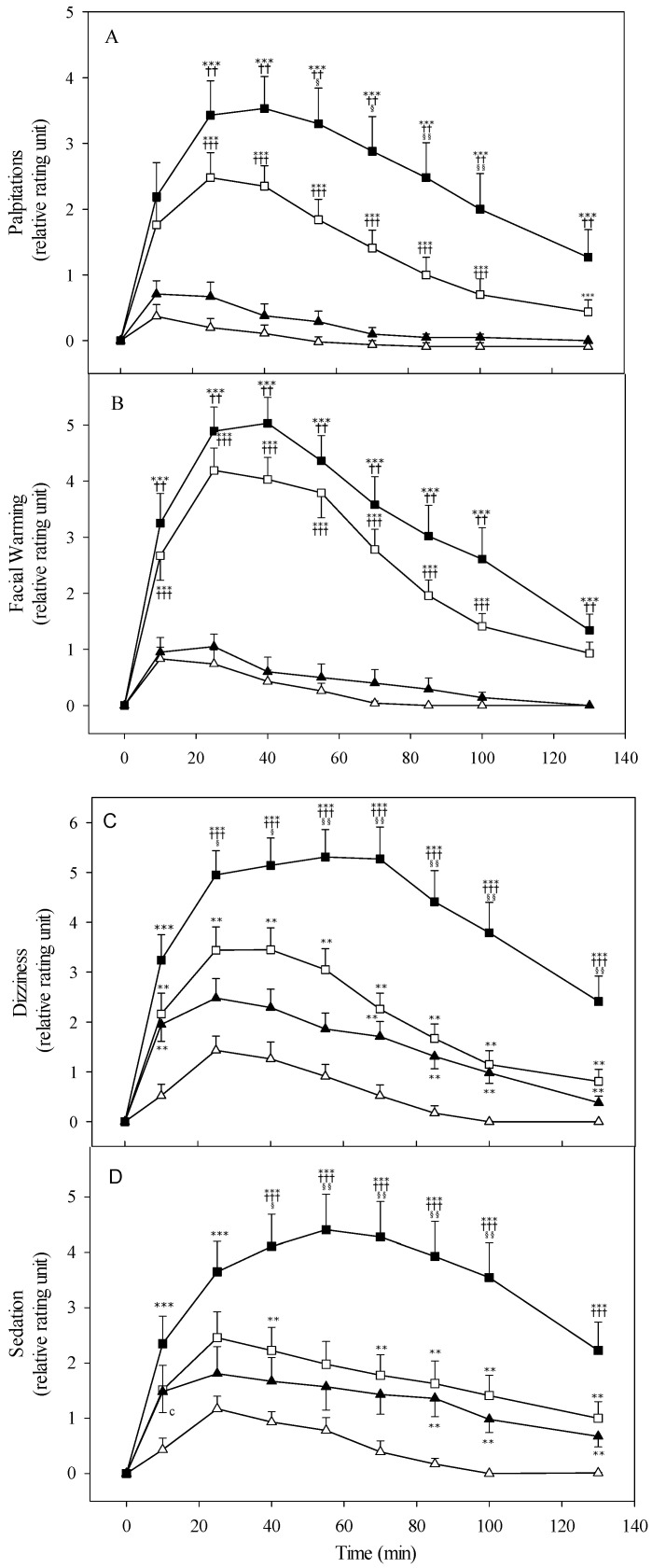
Alterations in subjective perceptions of palpitations (**A**), facial warming (**B**), dizziness (**C**), and sedation (**D**) of nonalcoholic subjects with different *ALDH2* allelotypes following different doses of ethanol (0.3 g/kg body weight or 0.5 g/kg). All subjects are *ADH1B*2/*2* homozygote. The study groups are denoted as △, *ALDH2*1/*1* with 0.3 g/kg ethanol (*n* = 23); ▲, *ALDH2*1/*1* with 0.5 g/kg ethanol (*n* = 21); □, *ALDH2*1/*2* with 0.3 g/kg ethanol (*n* = 27); ■, *ALDH2*1/*2* with 0.5 g/kg ethanol (*n* = 22). Vertical bars (for clarity only the upper or lower portion shown) represent standard errors of the mean. Statistically significant differences between groups at the corresponding time points was evaluated by ANOVA. *** *p* < 0.001, ** *p* < 0.01 vs. *ALDH2*1/*1* with 0.3 g/kg ethanol; ^+++^
*p* < 0.001, ^++^
*p* < 0.01 vs. *ALDH2*1/*1* with 0.5 g/kg ethanol; ^§§^
*p* < 0.01, ^§^
*p* < 0.05 vs. *ALDH2*1/*2* with 0.3 g/kg ethanol.

**Figure 5 biomolecules-11-01183-f005:**
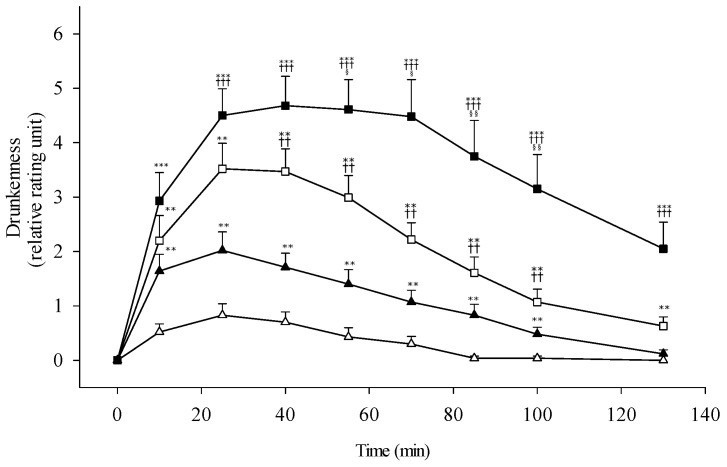
Feelings of drunkenness in nonalcoholic subjects with different *ALDH2* allelotypes following different doses of ethanol (0.3 g/kg body weight or 0.5 g/kg). All subjects are *ADH1B*2/*2* homozygote. The study groups are denoted as △, *ALDH2*1/*1* with 0.3 g/kg ethanol (*n* = 23); ▲, *ALDH2*1/*1* with 0.5 g/kg ethanol (*n* = 21); □, *ALDH2*1/*2* with 0.3 g/kg ethanol (*n* = 27); ■, *ALDH2*1/*2* with 0.5 g/kg ethanol (*n* = 22). Vertical bars (for clarity only the upper or lower portion shown) represent standard errors of the mean. Statistically significant differences between groups at the corresponding time points was evaluated by ANOVA. *** *p* < 0.001, ** *p* < 0.01 vs. *ALDH2*1/*1* with 0.3 g/kg ethanol; ^+++^
*p* < 0.001, ^++^
*p* < 0.01 vs. *ALDH2*1/*1* with 0.5 g/kg ethanol; ^§§^
*p* < 0.01, ^§^
*p* < 0.05 vs. *ALDH2*1/*2* with 0.3 g/kg ethanol.

**Figure 6 biomolecules-11-01183-f006:**
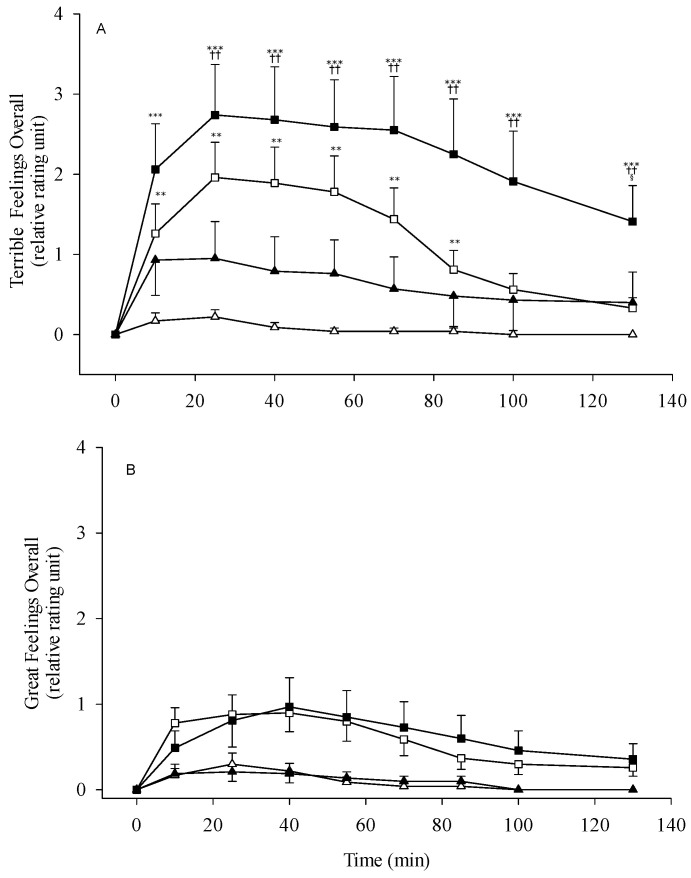
Overall terrible feelings (**A**) and great feelings (**B**) in nonalcoholic subjects with different *ALDH2* allelotypes following different doses of ethanol (0.3 g/kg body weight or 0.5 g/kg). All subjects are *ADH1B*2/*2* homozygote. The study groups are denoted as △, *ALDH2*1/*1* with 0.3 g/kg ethanol (*n* = 23); ▲, *ALDH2*1/*1* with 0.5 g/kg ethanol (*n* = 21); □, *ALDH2*1/*2* with 0.3 g/kg ethanol (*n* = 27); ■, *ALDH2*1/*2* with 0.5 g/kg ethanol (*n* = 22). Vertical bars (for clarity only the upper or lower portion shown) represent standard errors of the mean. Statistically significant differences between groups at the corresponding time points was evaluated by ANOVA. *** *p* < 0.001, ** *p* < 0.01 vs. *ALDH2*1/*1* with 0.3 g/kg ethanol; ^++^
*p* < 0.01 vs. *ALDH2*1/*1* with 0.5 g/kg ethanol; ^§^
*p* < 0.05 vs. *ALDH2*1/*2* with 0.3 g/kg ethanol.

**Table 1 biomolecules-11-01183-t001:** Characteristics of individuals with different *ADH1B* and *ALDH2* genotypes for subjective response study.

	*ADH1B*2/*2–ALDH2*1/*1*		*ADH1B*2/*2–ALDH2*1/*2*		*ADH1B*1/*1–ALDH2*1/*1*	
	**0.3 g/kg** **(*n* = 23)**	**0.5 g/kg** **(*n* = 21)**	**0.3 g/kg** **(*n* = 27)**	**0.5 g/kg** **(*n* = 22)**	**0.3 g/kg** **(*n* = 10)**	**0.5 g/kg** **(*n* = 10)**
Age (years)	22.6 ± 0.7	22.5 ± 0.6	23.1 ± 0.8	23.0 ± 0.8	22.9 ± 1.0	22.9 ± 1.0
Body weight (kg)	67.8 ± 7.2	68.2 ± 7.3	66.4 ± 6.8	66.6 ± 6.7	62.8 ± 5.6	62.8 ± 5.6
Body mass index (kg/m^2^)	22.8 ± 2.5	22.8 ± 2.5	22.0 ± 2.0	22.0 ± 2.0	21.3 ± 1.8	21.3 ± 1.8

Values are mean ± SEM. The subjects participated in the study received two different doses of ethanol (0.3 and 0.5 g/kg) in separate test sessions. No statistically significant differences within and between the combinatory genotype groups were found by analysis of variance (ANOVA).

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
