# Peer review of "Acetaldehyde Enhances Alcohol Sensitivity and Protects against Alcoholism: Evidence from Alcohol Metabolism in Subjects with Variant ALDH2*2 Gene Allele"

_biomolecules, 2021, doi:10.3390/biom11081183_

Round 1
Reviewer 1 Report
Title: Acetaldehyde Enhances Alcohol Sensitivity and Protects Against Alcoholism: Evidence from Alcohol Metabolism in Subjects with Variant ALDH2*2 Gene Allele.
This study is an alcohol challenge study with two dose sessions (0.3 g/kg and 0.5 g/kg) examining 60 healthy young males with combinations of different ADH1B and ALDH2 genotypes to investigate the association between alcohol sensitivity and ADH1B and ALDH2 genotypes.
The authors main findings were 1) dose-dependent increase in blood alcohol concentration regardless of ADH1B and ALDH2 genotypes; 2) significant build-up of blood acetaldehyde levels among subjects with ALDH2*2 allele, correlated with the dose of ingested alcohol; 3) significant increase in heart rate after alcohol ingestion were seen only among subjects with the ALDH2*2 allele along with subjective perceptions, such as facial warming, palpitation, dizziness, and sedation that were markedly elevated among subjects with the ALDH2*2 allele compared to that in those without the ALDH2*2 allele; and 4) no significant effect of ADH1B genotypes were found in alcohol metabolism or responses to alcohol. From these results, the authors concluded that subjective perceptions were primarily caused by acetaldehyde accumulation and acetaldehyde plays a major role in determining alcohol sensitivity, and thus, in protecting the development of alcohol dependence.
This study is well designed and has clearly shown the effect of acetaldehyde on subjective perceptions and an increase in heart rate. However, my major concern is that these findings have been reported previously by the authors’ group, and this article does not contain any new findings. The authors reported marked increase in heart rate, palpitations, facial warming, dizziness and terrible feelings among subjects with ALDH2*2 allele using the oral alcohol challenge test and concluded that acetaldehyde, rather than alcohol or acetate was primarily responsible for alcohol sensitivity reactions (Peng G-S, et al., Pharmacogenetics and Genomics, 2007).
The following are minor points.
Overall
There are many typographical errors with ADH written as AHD.
Introduction
Page 2, lines 54-55: the authors write that “Asians including Han Chinese, Japanese, and Koreans.” However, cited references examining Koreans are not included and an article examining Thai men is included.
Author Response
Regarding the reviewer’s concern about the differences of this manuscript and our prior publication (Peng G-S, et al., Pharmacogenetics and Genomics, 2007), there are different experiments designed to explore the pharmacokinetic and pharmacodynamic effect of alcohol metabolism. Our previous report focused on the responses of cardiac hemodynamic parameters and cranial arterial blood flow which were measured by two Doppler ultrasonography. As the setting of the hospital examination room is not naturalistic and it could influence the emotional state of the participants. The intermittent measurements of cardiovascular function during the test session might have restricted a detailed analysis of psychological response. As our previous publication already addressed the limitation and wrote “the subjective feelings reported here can be considered seen a preliminary investigation” in our previous article.
For different purpose, the study is separately designed to comprehensively investigate the alcohol perception based on previous preliminary results. In this study, we created a new Alcohol Response Scale and assessed the construct validity and reliability before the alcohol test. More eligible participants were recruited and the study procedure was conducted in relative naturalistic situation. The experiment integrated the pharmacokinetics and psychological perception to compare the ethanol/acetaldehyde effect on the alcohol sensitivity with two separate doses, 0.3 and 0.5 g/kg of ethanol. There are some important new findings in this paper as following:
- Dose effect on the blood ethanol and acetaldehyde concentration: The 1.7-fold increase of blood ethanol was highly correlated with 0.5 and 0.3 g/kg ethanol ingestion dose. In the participants with ALDH1*1/*2 heterozygote, peak blood acetaldehyde appeared to be 1.8-fold increase, 46.3 µM vs. 86.0 µM, following the 0.3 and 0.5 g/kg ethanol ingestion, respectively. To my knowledge, this is first report to systemically analyze the correlation of pharmacokinetic and dose effects in subjects carrying the different combinatory ADH1B and ALDH2
- The larger sample size allowed the correlation analysis between measurement parameters, ex. higher correlation between actual heart rate measurement and subjective feelings, up to 0.6 − 0.8. It implied good reliability and validity of Alcohol Response Scale to assess the physiological and psychological feelings after intaking different dose of ethanol. The correlation analyses provide the good evidence to prove the reliable questionnaire to assess the psychological responses to the effects of alcohol and acetaldehyde.
- Acetaldehyde has the synergistic effect to produce stronger psychological reaction and persistent augmentation of alcohol sensitivity in the ALDH1*1/*2 It provides powerful evidence to indicate the blood acetaldehyde can deter heavy alcohol drinking and prevent the development of alcoholism.
The typographic errors of AHD are corrected in the revised manuscript.
The reference 18# for Korean population was added and replaced the original Thai citation: Kim et al., Major genetic components underlying alcoholism in Korean population. Hum Mol Genet 2008, 17, 854-858.
Reviewer 2 Report
The authors report the result of a small experimental study of the effects of various genotypes on ethanol and acetaldehyde levels in blood following alcohol ingestion and research of various intoxication outcomes.
The conclusions are plausible and backed up by the necessary data.
The following revisions should be considered:
Abstract: add number of participants per group
Throughout: the English and terminology should be corrected by a native speaker
Introduction: can this be made more concise focused on the research question? This is quite long.
General comment: did you detect any endogenous acetaldehyde? There is still an ongoing debate about the level of endogenous acetaldehyde, if any.
Line 45: first pass or path?
Line 50: add space before The
Lines 57, 95: add space before reference number (check throughout)
Line 113-114: is there a rationale for the group sizes? Does the heterogeneity in the group sizes cause statistical problems?
Line 122-123: delete sentence, repetition of line 120
Line 141: add space before allelotypes
Line 173: body mass instead of max
All figures: what I do not understand is that the experimental design (Table 1) encompasses three types of individuals at 2 ethanol levels. However, there are only 4 lines shown in each figure. Should it not be 6 lines? Or is the active enzyme metabolizing so rapidly that all levels are zero? This should be clarified.
Line 246: delete space before comma
Line 336: add space before reference number
Can the full data be added as a data appendix?
Author Response
The revised manuscript is revised as the Reviewer pointing out. We respond to your helpful suggestion as following:
Abstract: add number of participants per group
As Reviewer suggestion, we add the number of participants in the abstract: ADH1B*2/*2–ALDH2*1/*1 (n = 23), ADH1B*2/*2–ALDH2*1/*2 (n = 27) and ADH1B*1/*1–ALDH2*1/*1 (n = 10) (Line 19-20).
Throughout: the English and terminology should be corrected by a native speaker
We are grateful for Editorial’s constructive suggestions. We rephrased some English sentences and corrected some writings problems as suggested by a native English speaker.
Introduction: can this be made more concise focused on the research question? This is quite long.
The introduction of revised manuscript contains 746 words in to describe the research background including the impacts of alcoholism, alcohol metabolic genes, alcohol sensitivity and research aims. It will be helpful for the readers who cited the article.
General comment: did you detect any endogenous acetaldehyde? There is still an ongoing debate about the level of endogenous acetaldehyde, if any.
As Reviewer points out, it is interesting issue for the endogenous acetaldehyde level in blood or in central nervous system. In this study, blood acetaldehyde was determined by high-performance liquid chromatography (HPLC), equipped with fluorescence detection and corrected for blank value. Based on our experimental finding, the acetaldehyde would increase by time in the HPLC analytic procedure. To avoid the time effect on the acetaldehyde production, the measurement values were corrected for the blank. Therefore, it is hard to answer the level of endogenous acetaldehyde addressed by Reviewer.
Line 45: first pass or path?
Be corrected as first-pass metabolism in revised manuscript (Line 46)
Line 50: add space before The,
Lines 57, 95: add space before reference number (check throughout)
Be corrected
Line 113-114: is there a rationale for the group sizes? Does the heterogeneity in the group sizes cause statistical problems?
Thank you for your notice. It is correct in the revised manuscript (Line 115). Corrected as: ADH1B*2/*2–ALDH2*1/*1 (n = 23), ADH1B*2/*2–ALDH2*1/*2 (n = 27) and ADH1B*1/*1–ALDH2*1/*1 (n = 10)
Line 122-123: delete sentence, repetition of line 120
Line 141: add space before allelotypes
Line 173: body mass instead of max
Be corrected as body mass index (Line 174)
All figures: what I do not understand is that the experimental design (Table 1) encompasses three types of individuals at 2 ethanol levels. However, there are only 4 lines shown in each figure. Should it not be 6 lines? Or is the active enzyme metabolizing so rapidly that all levels are zero? This should be clarified.
Thank you for Reviewer’s suggestion for presenting the results of ADH1B*1/*1–ALDH2*1/*1 group. The results of the pharmacokinetics and subjective perceptions were shown in the supplements (1-5). To compare the alterations of measurement parameters in nonalcoholic subjects with different ADH1B allelotypes following two separated different doses of ethanol (0.3 and 0.5 g/kg body weight), all subjects are ALDH2*1*1 homozygotes. The findings were also placed in the result part of text (Line 210, 231, 343 and 345 in Track Change version).
Line 246: delete space before comma
Line 336: add space before reference number
Be corrected
Can the full data be added as a data appendix?
The data are not publicly available due to existing ethnic and biological features for our population.
Reviewer 3 Report
In this study, the authors investigated the sensitivity to alcohol in 60 male volunteers screened for their combinatory genotypes AADH1B and ALDH2, and exposed to wo different doses of Ethanol (0.3 g/kg and 0.5/kg). THe resutls reported in the paper indicate that expose to alcohol resulted in an increase in EtOH and blood acetaldehyde in a dose dependent manner primarily in individuals with ALDH2*2 genotype. Consistent with this observatio, the volunteers with such a genotype experienced increased heart rate, dizziness, palpitations, and sense of drunkness as compared to other genotypically different volunteers. The conclusions of the authors is that ALDH2*2 plays a major role ofr acetaldehyde-related physiological negative responses and could represent a 'genetic' based protection against development of alcoholism in East Asians.
THe study appears properly conducted and the conclusions are in line with the results reported in the manuscript.
THe following points need to be clarified before the manuscript is acceptable for publication.
- Why were only male volunteers used for the study? A rationale should be provided in the Materials and Methods session
- The authors reports thorughout the REsults section that there were no significant differences between ADH1B*1*1 and ADH1B*2*2 with homozygous ALDH2*1 genotype. The authors, should indicate in the text that these results are not shown. Perhaps these results could be placed in 1 or 2 Tables as Supplemental data?
- English should be revised: several sentences are not syntactically correct, while others are unnecessarily repeated (e.g. line 120 and 122)
Author Response
- Some reports have demonstrated gender differences in alcohol metabolism and higher blood ethanol concentration after drinking equivalent amounts of The research goals are to investigate the alcohol metabolism and related subjective perceptions. To reduce the gender effect, we only recruited homogenous male from a total of 545 medical college male students who had been genotyped at the ADH1B and ALDH2 gene loci. The reason is described in the revised manuscript (Line152).
- Thank you for Reviewer’s suggestion for presenting the results of ADH1B*1/*1–ALDH2*1/*1 group. The results of the pharmacokinetics and subjective perceptions were shown in the supplements (1-5). To compare the alterations of measurement parameters in nonalcoholic subjects with different ADH1B allelotypes following two separated different doses of ethanol (0.3 and 0.5 g/kg body weight), all subjects are ALDH2*1*1 The findings were also placed in the result part of text (Line 210, 231, 343 and 345 in Track Change version).
- We rephrased some English sentences and corrected some writings problems as suggested.
Round 2
Reviewer 1 Report
Last time, I pointed out that findings presented in this article have been reported previously by the authors’ group, and this article does not contain any new findings. The authors were not responsive to the concerns that I raised last time. The authors should clearly state what are their new findings.
The points I noticed in this peer review are itemized below:
1) Detailed information of liver function on their subjects are lacking. A simple liver transaminases analysis might clarify the functional hepatic status and this could help to avoid this relevant factor on acetaldehyde production.
2) Did the authors collect data on history of alcohol exposure from their subjects? The history of alcohol use might impact the observed findings.
Minor comments:
The square symbols in supplement 5 is triangles in the figure legend.
Author Response
Comments and Suggestions for Authors
Last time, I pointed out that findings presented in this article have been reported previously by the authors’ group, and this article does not contain any new findings. The authors were not responsive to the concerns that I raised last time. The authors should clearly state what are their new findings.
Response to the Reviewer:
Many thanks for your comments on this manuscript regarding the concerned about the results reported in our previous publication. We would like to point out that there are clear differences between these two studies. This is a NEW study and a NEW set of experiments. The results from this study have NOT been reported.To respond Reviewer’s concern, we have highlighted these new features/aspects and revise our manuscript now as following:
- The experimental designs are different in exploring the pharmacokinetic and pharmacodynamic effect of alcohol metabolism in these two studies. Our previous report focused on the responses of cardiac hemodynamic parameters and cranial arterial blood flow which were measured by two Doppler ultrasonography. As the setting of the hospital examination room and intermittent measurements of cardiovascular function could influence the emotional state of the participants. The weakness of our previous study was not naturalistic and the report could be considered seen a preliminary investigation. We clarified the differences and added the sentences in last paragraph of Introduction (Line 98-103 in the revised manuscript).
- To address these weaknesses, in this study, we separately designed a very comprehensive measure of the alcohol perception and created a new Alcohol Response Scale and assessed the construct validity and reliability before this alcohol challenge study (described in Line 162-171).
- More eligible participants with relative homogenous characteristics were recruited and the study procedure was conducted in relative naturalistic situation. The experiment integrated the pharmacokinetics and psychological perception to compare the ethanol and acetaldehyde effect on the alcohol sensitivity with two separate doses, 0.3 and 0.5 g/kg of ethanol. These are all new and reported features and findings from this submitted manuscript (shown in Table 1, described in results as Line 193-197 and discussion as Line 324-330).
- We would like to highlight the new findings in this study as following:
(1). Dose-dependent effect on the alcohol metabolism: The same individual received two separate doses, 0.3 and 0.5 g/kg of ethanol. The study designs would mimic real drinking situation than our previous single dose experiment. The results allowed to further clarify the dose-dependent increase of ethanol concentration, ex. 1.7-fold increase of blood ethanol highly correlated with 0.5 and 0.3 g/kg ethanol ingestion dose (shown in Figure 1 and addressed in Line 212-217).
(2). Different level of acetaldehyde accumulation following separate ethanol dose ingestion: Individuals with ALDH1*1/*2 heterozygote, peak blood acetaldehyde appeared to be 1.8-fold increase, 46.3 µM vs. 86.0 µM, following the 0.3 and 0.5 g/kg ethanol ingestion, respectively. To my knowledge, this is first report to systemically analyze the correlation of pharmacokinetic and dose effects in subjects carrying the different combinatory ADH1B and ALDH2 genotypes (Shown in Figure 2 and highlighted the strong point in Line 336-338). 
(3). The larger sample size allowed the correlation analysis between measurement parameters, ex. higher correlation between actual heart rate measurement and subjective feelings, up to 0.6 − 0.8. It implied good reliability and validity of Alcohol Response Scale to assess the physiological and psychological feelings after intaking different dose of ethanol. The correlation analyses provide the good evidence to prove the reliable questionnaire to assess the psychological responses to the effects of alcohol and acetaldehyde (Line 270-274). 
(4). The results also demonstrated the diverse psychological reactions to the different blood level of ethanol and acetaldehyde, especially in the sensations of sedation and drunkenness. Acetaldehyde intensified the alcohol sensitivity during entire testing session and modulated by dose. Acetaldehyde has the synergistic effect to produce stronger psychological reaction and persistent augmentation of alcohol sensitivity in the ALDH1*1/*2 heterozygote. The findings prove our previous finding and provide powerful evidence to indicate the blood acetaldehyde can deter heavy alcohol drinking and prevent the development of alcoholism (shown in Figures 4-6, addressed in Line 399-403).
The points I noticed in this peer review are itemized below:
1) Detailed information of liver function on their subjects are lacking. A simple liver transaminases analysis might clarify the functional hepatic status and this could help to avoid this relevant factor on acetaldehyde production.
As the Reviewer points out the liver function and alcohol metabolism in liver, it is very important issue to influence the ethanol oxidation. In this study, the participants were students of National Defense Medical Center and received the routine healthy examination annually. Only the healthy subjects without the history of hepatitis and medical disease were recruited into the study (Line 120-122).
Reflecting on the results of alcohol pharmacokinetics, there are relative homogeneous ethanol concentration depending on ethanol ingesting dose (shown in Figure 1) and acetaldehyde accumulated only in the subjects with ALDH2*1/*2 heterozygote (shown in Figure 2).
2) Did the authors collect data on history of alcohol exposure from their subjects? The history of alcohol use might impact the observed findings.
As Reviewer addressed the history of alcohol drinking would influence the alcohol perception and sensitivity. In the study, we already excluded the subjects with problematic alcohol drinking and abuse history. The participant students lived on-campus which provided room and board. They were off-campus during weekends and summer/winter breaks, so they had relatively uniform lifestyle and nutritional status. None of the subjects drank alcoholic beverage more than occasionally. All participants were requested not to take alcohol for at least 1 week. It was partly corroborated by a blank level for both blood ethanol and acetaldehyde examined before the alcohol challenge test (shown in Line 124-132).
Minor comments:
The square symbols in supplement 5 is triangles in the figure legend.
Thank you very much for carefully reviewing our manuscript. The domain of psychological positive response was already shown in Supplement 4. For precise presentation, we would like to delete the Supplement 5 and edit the result part of revised manuscript (Line 311-314).
All changes are shown in the revised manuscript and marked up in the
“Track Changes” version
We greatly appreciate the Editor’s and Reviewer’s helpful comments.
Best regards,
Yi-Chyan Chen M.D., Ph.D.